Comparison of the recovery quality between remimazolam and propofol after general anesthesia: systematic review and a meta-analysis of randomized controlled trials

Zhu Caiyun 1 2
Xie Ran 2
Qin Fang 1 3
Wang Naiguo 4
Tang Hui tanghui1110@163.com 1 5
1 DepartmentofPharmacy, Shandong Provincial Hospital Affiliated to Shandong First Medical University , Jinan , China
2 DepartmentofPharmacy, Zibo Integrated Traditional Chinese and Western Medicine Hospital , Zibo , China
3 DepartmentofPharmacy, Liuzhou People’s Hospital , Guangxi , China
4 Department of Spinal Surgery, Shandong Provincial Hospital Affiliated to Shandong First Medical University , Jinan , China
5 Stem Cell Clinical Institute, Shandong Provincial Hospital Affiliated to Shandong First Medical University , Jinan , China
Albuquerque Ulysses
Electronic publication date: 2024 Aug 26
Publication date: 2024
Volume: 12
Electronic Location ID: e17930
Received 2024 Apr 30; Accepted 2024 Jul 25
Copyright: ©2024 Zhu et al.
Copyright year: 2024
Copyright holder: Zhu et al.
License: This is an open access article distributed under the terms of the Creative Commons Attribution License, which permits unrestricted use, distribution, reproduction and adaptation in any medium and for any purpose provided that it is properly attributed. For attribution, the original author(s), title, publication source (PeerJ) and either DOI or URL of the article must be cited.
License URL: https://creativecommons.org/licenses/by/4.0/

Keywords: General anaesthesia, Meta-analysis, Propofol, Quality of recovery, QoR, Remimazolam, Systematic review

Funding: National Natural Science Foundation of China 82102372 Natural Science Foundation of Shandong Province No. ZR2023MH264 Zibo Medical and Health Research Project 20231901134 This research was funded by the National Natural Science Foundation of China (82102372); Natural Science Foundation of Shandong Province (No. ZR2023MH264); Zibo Medical and Health Research Project (20231901134). The funders had no role in study design, data collection and analysis, decision to publish, or preparation of the manuscript.

==============================
Introduction

To evaluate the recovery quality between remimazolam and propofol after general anesthesia surgery.

Methods

We included eligible randomized controlled trials (RCTs) in EMBASE, PubMed, Cochrane Central, Scopus, and Web of Science up to June 26, 2024 for comparison the recovery quality of remimazolam and propofol after general anaesthesia. The primary outcomes were the total Quality of Recovery-15 (QoR-15) and five dimensions of QoR-15 on postoperative day 1 (POD1). Secondary outcomes were adverse events, the Quality of Recovery-40 (QoR-40) on POD1, and the intraoperative and postoperative time characteristics.

Results

Thirteen RCTs with a total of 1,305 patients were included in this meta-analysis. Our statistical analysis showed that remimazolam group had higher QoR-15 score on POD1, with no significant difference (Mean Difference (MD) = 1.24; 95% confidence interval (CI), [−1.67–4.15]; I2 = 75%; P = 0.41). In the five dimensions of QoR-15, remimazolam group was superior to propofol group in terms of physical independence (MD = 0.79; 95% CI [0.31–1.27]; I2 = 0%; P = 0.001). Remimazolam group was lower than propofol group in incidence of hypotension (Risk Ratio (RR) = 0.48; 95% CI [0.40–0.59]; I2 = 14%; P < 0.00001), bradycardia (RR = 0.18; 95% CI [0.08–0.38]; I2 = 0%; P < 0.0001) and injection pain (RR = 0.03; 95% CI [0.01–0.12]; I2 = 48%; P < 0.00001), respectively. The intraoperative and postoperative time characteristics and the QoR-40 were similar in the two groups.

Conclusions

Our analysis showed that the recovery quality of the remimazolam group after general anaesthesia was similar to propofol group, while the incidence of adverse events was low in remimazolam group. As a potential anesthetic, remimazolam can be used in place of propofol for surgical general anesthesia.

Introduction

Surgical technology has unique advantages in clinic treatment and anesthetic plays an important role in surgical techniques. Propofol has long been considered a more comfortable administration of general anesthesia than inhalation anesthesia because of its low incidence of nausea and vomiting. However, it also has limitations, such as low blood pressure and a high incidence of injection pain (Keam, 2020). Remimazolam is a newly benzodiazepine sedative/anesthesia that can be administered intravenous (Kilpatrick, 2021). Due to rapid metabolism by tissue esterases into inactive metabolites (Lee & Shirley, 2021), it has the characteristics of rapid onset and offset in vivo, and can be antagonized by flumazenil (Ustundag, Karasu & Urun, 2021). Clinical trials have demonstrated the safety and efficacy of general anesthesia (Ko et al., 2023). Remimazolam has a similar sedative effect to propofol and is superior to propofol in terms of injection pain and the risk of hypotension. Acturally, propofol may be superior to remimazolam in terms of depth of anesthesia (Zhang et al., 2022). However, the comparison between remimazolam and propofol in postoperative recovery quality is controversial.

In recent years, doctors and patients are increasingly concerned about the quality of recovery (QoR) after general anesthesia, not just the success or failure of surgery (Mao et al., 2022; Wessels et al., 2022). QoR is a broad concept that assesses recovery from multiple perspectives of the patient (Royse, 2017), many factors can affect the quality of postoperative recovery, such as pain, stress reaction, cognitive disorder, physical dysfuction, and emotional state (Bowyer & Royse, 2016). QoR-15 is the most widely used to evaluate the quality of postoperative recovery and comprehensive assessment from five dimensions (physical comfort, physical independence, emotional state, psychological support, and pain), the validity and reliability of QoR-15 has been verified in 16 countries and 15 languages (Myles et al., 2022). The QoR-15 was evolved from the larger QoR-40, and they were just equally effective in measuring quality of postoperative recovery (Gornall et al., 2013).

Choosing the right drug among the various anesthetics to improve QoR can be a challenge. A meta-analysis of the hemodynamic effects by Peng et al. (2023) only included two RCTs that showed no significant difference between remimazolam and propofol in total QoR-15 scores on POD1, as a result, conclusion was low reliability and cannot be applied to more types of surgery. We systematically updated this study by collecting 13 RCTs and conducted a comprehensive meta-analysis about quality of recovery after general anaesthesia between remimazolam and propofol.

Methods

We conducted and reported analyses in accordance with the Preferred Reporting Items for Systematic Reviews and Meta-Analyses (PRISMA) 2020. The protocol has been listed in PROSPERO International Prospective Register of Systematic Reviews (CRD42024497497).

Search strategy

We searched EMBASE, PubMed, Scopus, Cochrane Central, and Web of Science databases for eligible studies up to June 26, 2024, in unrestricted languages. Checking the registration number at Clinical Trials and the database to make sure no data was missing. The search strategy is as follows: “Remimazolam*” AND “Propofol” [Mesh] OR “Propofol*” AND “Randomized controlled trial” OR “Randomized” OR “Randomly” OR “random” AND “quality of recovery” OR “recovery quality” OR “QoR”, and we did not search grey literature. The search strategy is shown in supplemental Table S1. CZ and FQ independently performed the search strategy and resolved their disagreements through discussion.

Eligibility criteria and study selection

The two authors completed the selection of articles independently. If there was different point of view, they resolved it through discussion. There are no language restrictions for articles in this search. We included all RCTs that met the following PICO (Population, Intervention, Comparator, Outcomes) study question criteria. (1) Population: adult patients (age ≥ 18) requiring surgery under general anesthesia, no surgical type is considered including cardiac surgery; (2) intervention: the study experimental group used remimazolam to induce and maintain anesthesia; (3) comparator: the study control group used propofol to induce and maintain anesthesia; other sedatives, muscle relaxants, analgesics, antiemietics, and nerve block techniques could be used in experimental and control groups. (4) Outcomes: quality of recovery index, intraoperative and postoperative time characteristic indexs and adverse events. We excluded studies with the following criteria: duplicate literature, review or meta-analysis articles, protocol, non-general anesthesia, non-randomized controlled trials, studies without using propofol as a control group, data unavailable for analysis, uneligible anesthetic strategy and no project data reported.

Outcome measures

The primary outcome was the recovery quality of total QoR-15 and five dimensions of QoR-15 on POD1 (physical comfort, physical independence, emotional state, psychological support, and pain) between remimazolam and propofol group. Secondary outcomes were duration of post-anesthesia care unit (PACU) stay, time to extubation, duration of anesthesia, duration of operation, duration of postoperative hospital stay, the QoR-40 and adverse events. The QoR-15 consists of 15 items, including physical comfort (five items), emotional state (four items), psychological support (two items), physical independence (two items) and pain (two items), each item is scored on an 11-point scale according to the frequency on the scale, the total score range from 0 to 150, the higher the score, the better the quality of recovery. The QoR-40 scale ranges from 40 to 200 and the higher the score, the better the quality of recovery.

Data extraction

The extracted data includes first author, year of publication, registration number, country, sample size, American Society of Anesthesiologists (ASA) physical status, body mass index (BMI), gender ratio, type of surgery, specific interventions and comparisons methods, scale design and predetermined outcomes. When the experimental groups based on different doses of remimazolam, we extracted the dose commonly used in this type of surgery. Two independent authors used standard tables for data extraction and resolved their disagreements through discussion.

Risk of bias assessment

Our two authors independently assessed the risk of bias domains using the Cochrane risk of bias assessment tool. The Cochrane risk of bias tool detects the following types of bias: random sequence generation, allocation concealment, binding of participants and personnel, blinding of outcome assessment, incomplete outcome date and selective reporting. Each bias domain was judged as having a high, unclear, or low risk of bias.

Statistical analysis

Review Manager (Rev Man 5.4.1; Cochrane Training, https://training.cochrane.org/online-learning/core-software/revman) was used in this meta-analysis, we pooled continuous outcomes as mean differences (MDs) with 95% CIs under the fixed-effect model. Inverse Variance method was utilized to calculate the MDs value. If the results were represented in quartiles, we contact the article author by email or phone to get the original data, and if there was no response, we convert the data to mean and standard deviation when it meets the conversion criteria. Moreover, Mantel-Haenszel risk ratio (RR) and 95% CI were used for the number of events and samples of dichotomous data. We adopted a 2-tailed test and P < 0.05 for the overall effect observed was indicated significant differences. Statistical heterogeneity was assessed with the Q test and I-square (I2) statistic test. The random effects model was used to evaluate the stability of the combined results of the fixed effects model. If a strong heterogeneity (I2 ≥50%) was found, a leave-one-out sensitivity analysis was employed to evaluate the single comparison-driven conclusion, and if no source of heterogeneity was found, a random effects model was put to use.

Results

Study results

A total of 249 relevant literatures were searched in EMBASE, PubMed, Scopus, Cochrane Central, and Web of Science databases Fig. 1. Thirty-six duplicate articles were removed, 169 were removed by reading the title and abstract, and 31 were removed after reading the full text. We also checked ClinicaTrials. gov to make sure there are no missing articles. Finally, we received 13 unique literatures were retrieved for this systematic review and meta-analysis (Chen et al., 2024; Choi et al., 2022; Huang et al., 2023; Jiao et al., 2024; Kim et al., 2023; Lee et al., 2023; Lee et al., 2024c; Lee et al., 2024a; Lee et al., 2024b; Luo et al., 2023a; Tang et al., 2023; Xiao et al., 2024; Zhao et al., 2023).

Figure 1 PRISMA 2020 flow diagram of articles.

Demographics and characteristics

Our analysis included thirteen studies with a total of 1305 patients, of which seven were conducted in China and six were conducted in Korea. Patients aged from 18 to 86 years were divided into remimazolam group and propofol group, respectively. The doses in the remimazolam group were not exactly the same, induction dose of remimazolam: seven RCTs used the dosage recommended by the instructions (6 mg/(kg h)), three RCT used larger dose (12 mg/(kg h)) and five RCTs used a bolus dose (0.2–0.3 mg/kg, 0.2 mg/kg and 0.3 mg/kg, respectively); maintenance dose of remimazolam: twelve RCTs used the dosage recommended by the instructions, only one RCT used a lesser dose (0.3 mg/(kg h)). Table 1 shows the baseline summary of the included RCTs.

Table 1 Baseline summary of the included RCTs.

Rank	Trails	Country	Study design	Sample size	ASA	Surgery type	Participant characteristics	Remimazolam	Propofol	Scale design	
							Remimazolam group	Propofol group				
1	Choi et al., 2022
(NCT05016518)	Korea	RCT	139	I, II	Open thyroidectomy	Age (39.5 (33–48))
F/M (70/0)	Age (41.0 (37–47))
F/M (69/0)	(IV)
Induction: 6 mg/(kg h)
Maintenance: 1–2 mg/(kg h)	(TCI)
Induction: 5 μg/mL
Maintenance: 2–6 μg/mL	QoR-15	
2	Lee et al., 2023
(NCT05047939)	Korea	RCT	57	I, II	Open thyroidectomy	Age (45 ± 13.4)
F/M (21/7)
BMI (24.3 (22.8–26.0))	Age (51 ± 12.1)
F/M (19/10)
BMI (22.6 (20.9–25.3))	(IV)
Induction: 6 mg/(kg h)
Maintenance: 1–2 mg/(kg h)	(TCI)
Induction: 3 ng/ml
Maintenance: 2 ng/ml	QoR-15	
3	Tang et al., 2023
(ChiCTR2100053014)	China	RCT	114	I, II	Meniscus repair	Age (48.5 (19–62))
F/M (29/27)
BMI (24.73 ± 2.93)	Age (50 (19–64))
F/M (31/27)
BMI (23.95 ± 2.85)	(IV)
Induction: 6 mg/(kg h)
Maintenance: 0.4–2 mg/(kg h)	(TCI)
Induction: 2 μg/mL
3.5 μg/mL after 20 s
Maintenance: 1–3 μg/ml	QoR-15	
4	Zhao et al., 2023	China	RCT	108	I, II	Esophagectomy	Age (65.4 ± 3.1)
F/M (19/35)
BMI (21.2 ± 0.8)	Age (64.5 ± 3)
F/M (21/33)
BMI (21.5 ± 0.8)	(IV)
Induction: 0.2–0.3 mg/kg
Maintenance: 0.4–1 mg/(kg h)	(IV)
Induction: 1–2 mg/kg
Maintenance: 4–10 mg/(kg h)	QoR-15	
5	Jiao et al., 2024
(ChiCTR2300068097)	China	RCT	90	I, II	Vocal cord polypectomy	Age (41.2 ± 10.9)
F/M (20/25)
BMI (24.2 ± 2.1)	Age (40.2 ± 9.1)
F/M (18/27)
BMI (23.8 ± 1.9)	(IV)
Induction: 0.2–0.3 mg/kg
Maintenance: 0.5–1 mg/(kg h)	(IV)
Induction: 1.2–2.0 mg/kg
Maintenance: 4–6 mg/(kg h)	QoR-15	
6	Lee et al., 2024a
(NCT05435911)	Korea	RCT	63	I, II	Breast cancer surgery	Age (54 ± 8)
F/M (32/0)
BMI (23.7 ± 3.2)	Age (54 ± 10)
F/M (31/0)
BMI (24.1 ± 4)	(IV)
Induction: 6 mg/(kg h)
Maintenance: 1–2 mg/(kg h)	(TCI)
Induction: 4 μg/mL
Maintenance: ≤ 4 μg/mL	QoR-15	
7	Lee et al., 2024b
(NCT05397886)	Korea	RCT	53	NA	Radiofrequency
catheter ablation	Age (66 (61–69))
F/M (6/20)
BMI (24.6 ± 3.8)	Age (58 (48–66))
F/M (5/22)
BMI (26.1 ± 2.8)	(IV)
Induction: 6 mg/(kg h)
Maintenance: 1–2 mg/(kg h)	(TCI)
Induction: 3–4 μg/mL
Maintenance: 3–4 μg/mL	QoR-15	
8	Lee et al., 2024c
(NCT04994704)	Korea	RCT	72	I, II, III	Spine surgery	Age (54.2 (27–66))
F/M (17/19)	Age (50.3 (33–66))
F/M (13/23)	(IV)
Induction: 6–12 mg/(kg h)
Maintenance: 1–2 mg/(kg h)	(TCI)
Induction: 3 ng/ml
Maintenance: 3 ng/ml	QoR-15	
9	Xiao et al., 2024
(ChiCTR2100049314)	China	RCT	84	I, II, III	Cholangiopan- creatography	Age (53 ± 12)
F/M (18/24)
BMI (20.5 ± 4.0)	Age (57 ± 13)
F/M (16/26)
BMI (21.3 ± 2.7)	(IV)
Induction: 0.2 mg/kg
Maintenance: 0.5–2 mg/(kg h)	(IV)
Induction: 1.5 mg/kg
Maintenance: 2–8 mg/(kg h)	QoR-15	
10	Kim et al., 2023
(KCT0006965)	Korea	RCT	189	I, II	Oral and maxillofacial
surgery	Age (41.7 ± 12.2)
F/M (33/61)
BMI (23.8 ± 3.3)	Age (43.3 ± 13.2)
F/M (36/59)
BMI (23.7 ± 2.9)	(IV)
Induction: 12 mg/(kg h)
Maintenance: 1–2 mg/(kg h)	(TCI)
Induction: 3–5 μg/mL
Maintenance: 3–5 μg/mL	QoR-40	
11	Huang et al., 2023
(ChiCTR2000040579)	China	RCT	120	II, III	Breast cancer surgery	Age (62.6 ± 8.9)
BMI (24.3 ± 2.6)	Age (63.8 ± 11)
BMI (24.8 ± 2.7)	(IV)
Induction: 0.3 mg/kg
Maintenance: 0.3 mg/(kg h)	(IV)
Induction: 2 mg/kg
Maintenance: 2 mg/(kg h)	QoR-40	
12	Chen et al., 2024
(ChiCTR2100053141)	China	RCT	108	I, II, III	Sleeve gastrectomy	Age (28.5 (23–33.3))
F/M (30/24)
BMI (41.6 (40.8–43.4))	Age (32 (28–34))
F/M (32/22)
BMI (43 (41.1–44.9))	(IV)
Induction: 0.2 mg/kg
Maintenance: 0–1 mg/(kg h)	(IV)
Induction: 1.5–3 mg/kg
Maintenance: 0–12 mg/(kg h)	QoR-40	
13	Luo et al., 2023a
(ChiCTR2000038094)	China	RCT	96	I, II	Laparoscopic surgery	Age (38.8 ± 13.3)
F/M (15/32)
BMI (23.7 ± 3.2)	Age (37.3 ± 12.1)
F/M (26/23)
BMI (23 ± 2.9)	(IV)
Induction: 6 mg/(kg h)
Maintenance: 1 mg/(kg h)	(IV)
Induction: 2 mg/kg
Maintenance: 6 mg/(kg h)	QoR-40	
Notes.

RCT randomized controlled trials

ASA American Society of anesthesiologists

BMI body mass index

IV intravenous

TCI target-controlled infusion

Quality assessment of included studies

According to the Cochrane Risk of Bias Assessment Tool, the quality of the included randomized controlled trials was estimated to be medium to high. The summary and risk of bias for the included studies are shown in Fig. 2.

Figure 2 Risk of bias assessment.

(A) Risk of bias evaluated by the Cocharne Collaboration Risk of Bias Assessment Instrument. (B) Risk of bias assessment for included articles.

Comparison of remimazolam with propofol in the term of total QoR-15 on the POD1

The primary outcome was the change in QoR-15 on the POD1. Four studies (Choi et al., 2022; Jiao et al., 2024; Lee et al., 2023; Zhao et al., 2023) with 370 patients examined QoR-15 on the pre-operation. Our statistical analysis showed there was no significant difference between the two groups (MD = −0.62; 95% CI [−1.83–0.59]; I2 = 37%; P = 0.31) (Fig. 3A). Nine studies (Choi et al., 2022; Jiao et al., 2024; Lee et al., 2023; Lee et al., 2024c; Lee et al., 2024a; Lee et al., 2024b; Tang et al., 2023; Xiao et al., 2024; Zhao et al., 2023) involving 780 patients investigated QoR-15 on the POD1. Our statistical analysis showed that the score of QoR-15 in remimazolam group was higher than propofol group, with no significant difference (MD = 1.24; 95% CI [−1.67–4.15]; I2 = 75%; P = 0.41), which indicated that remimazolam group and propofol group had similar quality of recovery on the POD1 (Fig. 3B). Due to the high heterogeneity, we adopted the leave-one-out method to eliminate the study of Zhao et al. (2023), and the analysis still reached the same conclusion (MD = 0.20; 95% CI [−2.53–2.94]; I2 = 63%; P = 0.88) (Fig. 3C).

Figure 3 Forest plot comparing between remimazolam group and propofol group.

(A) QoR-15 on the pre-operation; (B) QoR-15 on the POD1; (C) QoR-15 after leave-one-out on the POD1 (CI, confidence interval; IV, inverse variance).

Comparison of remimazolam with propofol in the term of five dimensions of QoR-15

We analyzed five dimensions of QoR-15 scores, emotional status, physical comfort, psychological support, physical independence, pain. Three studies (Choi et al., 2022; Tang et al., 2023; Zhao et al., 2023) involving 361 patients have reported these five dimensions. In term of physical independence, remimazolam group was better than propofol group, with significant difference and low heterogeneity (MD = 0.79; 95% CI [0.31–1.27]; I2 = 0%; P = 0.001) (Fig. 4D). There was no significant difference in emotional status, physical comfort, psychological support and pain between remimazolam group and propofol group (p > 0.05) (Figs. 4A, 4B, 4C, 4E).

Figure 4 Forest plot comparing between remimazolam group and propofol group.

(A) Emotional status (B) physical comfort (C) psychological support (D) physical independence (E) pain (CI, confidence interval; IV, inverse variance).

Comparison of remimazolam with propofol in the term of intraoperative and postoperative time characteristics and QoR-40 on POD1

We also analyzed other factors related to the quality of postoperative recovery. Ten (Chen et al., 2024; Choi et al., 2022; Jiao et al., 2024; Lee et al., 2023; Lee et al., 2024c; Lee et al., 2024a; Lee et al., 2024b; Luo et al., 2023a; Tang et al., 2023; Zhao et al., 2023), eight (Choi et al., 2022; Kim et al., 2023; Lee et al., 2024c; Lee et al., 2024b; Luo et al., 2023a; Tang et al., 2023; Xiao et al., 2024; Zhao et al., 2023), 10 (Chen et al., 2024; Jiao et al., 2024; Kim et al., 2023; Lee et al., 2023; Lee et al., 2024c; Lee et al., 2024a; Luo et al., 2023a; Tang et al., 2023; Xiao et al., 2024; Zhao et al., 2023), 10 (Chen et al., 2024; Choi et al., 2022; Jiao et al., 2024; Lee et al., 2023; Lee et al., 2024c; Lee et al., 2024a; Luo et al., 2023a; Tang et al., 2023; Xiao et al., 2024; Zhao et al., 2023), three (Choi et al., 2022; Lee et al., 2023; Zhao et al., 2023) studies were involved in time to extubation, duration of anesthesia, duration of surgery, duration of PACU and duration of postoperative hospital stay, respectively. However, no significant differences were observed in these respects (p > 0.05), and the heterogeneity was low to high (Figs. 5A–5E). The QoR-40 scale is equivalent to the QoR-15 scale in evaluating postoperative recovery quality. Four (Chen et al., 2024; Huang et al., 2023; Kim et al., 2023; Luo et al., 2023a) studies were involved QoR-40 on the POD1, no significant differences were observed in QoR-40 (p > 0.05), and the heterogeneity was low (Fig. 5F), which indicated that remimazolam group and propofol group had similar quality of recovery on the POD1.

Figure 5 Forest plot comparing between remimazolam group and propofol group.

(A) Time to extubation; (B) duration of annesthesia; (C) duration of surgery; (D) duration of PACU; (E) duration of postoperative hospital stay; (F) QoR-40 on the POD1 (CI, confidence interval; IV, inverse variance).

Incidence of adverse events

Table 2 shows adverse events reported by two or more RCTs. The remimazolam group was lower than the propofol group in incidence of hypotension (RR = 0.48; 95% CI [0.40–0.59]; I2 = 14%; P < 0.00001), bradycardia (RR = 0.18; 95% CI [0.08–0.38]; I2 = 0%; P < 0.0001) and injection pain (RR = 0.03; 95% CI [0.01–0.12]; I2 = 48%; P < 0.00001) and the difference was significant and low heterogeneity. There was no significant difference in the incidence of postoperative nausea/vomiting (PONV) between the two groups. (Fig. S1).

Table 2 Adverse events.

References	Sample size	PONV, n	Injection pain, n	Hypotension, n	Bradycardia, n	
	R	P	R	P	R	P	R	P	R	P	
Choi 2022	70	69	NA	NA	0	2	1	7	NA	NA	
Lee 2023	28	29	3	0	NA	NA	NA	NA	NA	NA	
Tang 2023	56	58	6	5	0	3	18	34	3	10	
Zhao 2023	54	54	NA	NA	NA	NA	19	34	3	18	
Kim 2023	94	95	11	10	NA	NA	NA	NA	NA	NA	
Huang 2023	60	60	2	1	NA	NA	22	35	NA	NA	
Jiao 2024	45	45	5	3	3	38	11	29	NA	NA	
Lee 2024a	32	31	6	4	NA	NA	NA	NA	NA	NA	
Lee 2024b	26	27	1	3	NA	NA	3	1	NA	NA	
Lee 2024c	36	36	13	12	NA	NA	NA	NA	NA	NA	
Xiao 2024	42	42	2	5	NA	NA	6	17	NA	NA	
Chen 2024	54	54	NA	NA	NA	NA	13	31	1	12	
Luo 2023	47	49	4	8	NA	NA	4	15	NA	NA	
Total incidence (%)			11.4	10.8	1.7	25	21.4	44.3	4.3	24.1	
Notes.

R: remimazolam group; P: propofol group; PONV: postoperative nausea/vomiting.

Discussion

In this meta-analysis study, we reviewed thirteen RCTs analyzing remimazolam for the quality of recovery after general anaesthesia. A total of 1,305 patients in remimazolam group and propofol group were included in our report. Our aim was to evaluate the recovery quality between remimazolam and propofol after general anesthesia surgery.

Remimazolam, used for general anesthesia in surgery, is a benzodiazepine with the basic sedative structure of midazolam and the ester structure of remifentanil. After entering the body, it directly binds to gamma-aminobutyric acid-a (GABAA) receptors producing anesthetic effect (Brohan & Goudra, 2017), with an onset time of 1–3 min. Tissue esterases in vivo can break the ester bond and metabolize it into inactive CNS7054 with lose efficacy time of 6.8–9.9 min (Kilpatrick, 2021). These pharmacokinetics characteristics enable the remimazolam to reach the operable state quickly, and patient can recover quickly after the surgery with few adverse reactions. Among the RCTs included in our analysis, seven RCTs used 6 mg/(kg h) for anesthesia induction and two RCTs used 12 mg/(kg h) or a bolus dosage. The use of large doses was based on the effectiveness and safety of previous studies (Chen et al., 2020; Doi et al., 2020).

According to our findings, the quality of recovery of remimazolam group was similar to propofol group, and that incidence of adverse effects, such as hypotension, bradycardia and injection pain, remimazolam group was lower than propofol group. In terms of time to extubation, duration of anesthesia, duration of surgery, duration of PACU and duration of postoperative hospital stay, remimazolam group was similar to propofol group.

At present, the scales commonly used to evaluate the quality of postoperative recovery between remimazolam and propofol include QoR-15, QoR-40 and the post-operative quality of recovery scale (PostopQRS).

QoR-15 was developed from QoR-40, and both were evaluated in five dimensions, including physical comfort, physical independence, emotional state, psychological support, and pain, with the highest scores were 150 and 200, respectively (Gornall et al., 2013; Stark, Myles & Burke, 2013). The higher score indicates a better recovery quality of rehabilitation (Myles, 2018). The RCT of Mao et al. (2022) evaluated QoR-15 after general anesthesia with remimazolam in urological surgery, showing that remimazolam group was significantly lower than propofol group. The meta-analysis by Peng et al. (2023) found no significant difference between remimazolam and propofol including Mao et al. (2022) and an RCT. We updated the data with eight RCTs showed a conclusion consistent with Peng et al. (2023). This may be related to similar sedation success rate (Chang et al., 2023; Zhang et al., 2022), time to extubation, duration of anesthesia and duration of surgery between remimazolam and propofol. After the data analysis, we found an interesting thing that the high heterogeneity of the total QoR-15 might be due to the inclusion of patients only aged 60 to 80 years in the RCT (Zhao et al., 2023); they concluded that remimazolam has a significantly higher superiority in postoperative recovery quality on the POD1. Previous meta-analyses showed that remimazolam has more stable hemodynamics than propofol, with lower incidence of hypotension, bradycardia, and respiratory depression (Ko et al., 2023; Wu et al., 2023), which are important to the elderly, and the advantages of remimazolam in anesthesia in the elderly might be the direction of future research. Another RCT in Japan comparing the hemodynamics of remimazolam and propofol showed no significant difference in QoR-15 on POD3 and a lower incidence of hypotension in remimazolam (Kotani et al., 2024).

We further wanted to validate previous results with QoR-40 on POD1 and concluded that there was no significant difference between QoR-40 in the remimazolam group and the propofol group, which also confirmed our conclusion based on QoR-15. The RCT of Li et al. (2021) used QoR-40 to evaluate the recovery quality in elderly patients on POD3, and found that remimazolam was significantly better than propofol. Two RCTs used PostopQRS scale enrolled colonoscopy patients aged 18 to 75 years, one research showed that remimazolam was superior to propofol in the rate of cognitive recovery on POD1 and postoperative day 7 (POD7) and the overall recovery rate on POD7 (Guo et al., 2022), while the other research showed that there was no significant difference between remimazolam and propofol in recovery quality on POD1 (Luo et al., 2023b). Another RCT used PostopQRS scale found that remimazolam provided a similar postoperative recovery quality to propofol at discharge (Zhang et al., 2024).

We also analyzed other measures related to QoR, which showed no significant differences in PACU residence time, extubation time, anesthesia time, surgical time, and postoperative hospital stay between remimazolam and propofol.

The adverse reactions of anesthetics should also be paid attention to. According to our statistical results, the incidence of injection pain, hypotension, and bradycardia in the remimazolam group was lower than those in the propofol group, which was consistent with the precious literature results (Ko et al., 2023; Zhang et al., 2022). It seems to indicate that remimazolam could be used as anesthesia in clinical surgery to better improve patient comfort.

This article presents the first comprehensive systematic review and meta-analysis of the recovery quality after general anesthesia surgery with remimazolam. There are some limitations to our study. The races included in our analysis were all Asian, so our results may only apply to Asians. We searched articles on remimazolam for non-Asians and actually there were no studies on racial subgroup analysis in clinical trials. A meta-analysis comparing the efficacy and safety of remimazolam with other sedatives, including six RCTs from China and five RCTs from the United States, was not performed for racial subgroup analysis (Tang et al., 2022). Another meta-analysis comparing the safety and efficacy of remimazolam and midazolam for endoscopic sedation, including five RCTs from the United States and two RCTs from China, was not performed the racial subgroup analysis either (Zhu et al., 2021). The age range of patients included in each RCT is different. It is not possible to determine whether there is a difference in the quality of postoperative recovery between elderly and non-elderly people. In the future, we will need larger sample sizes for subgroup analysis.

Conclusions

Our systematic review and meta-analysis showed that in patients with general anaesthesia surgery, remimazolam treatment was similar to propofol group in postoperative recovery quality; however, the former has a lower incidence of injection pain, hypotension and, bradycardia. Remimazolam comparable to propofol group in terms of intraoperative and postoperative time characteristics. More RCTs with larger sample sizes and longer follow-up periods are needed to consolidate the benefits of recovery quality in patients treated with remimazolam.

Supplemental Information

Supplemental Information 1 The complete search strategy of PubMed

Supplemental Information 2 Forest plot comparing the risk of (A) Injection pain (B) Hypotension (C) Bradycardia (D) postoperative nausea/vomiting (PONV) between remimazolam and propofol groups. M-H, Mantel-Haenszel; CI, confidence interval

M-H, Mantel-Haenszel; CI, confidence interval.

Supplemental Information 3 PRISMA 2020 checklist

Additional Information and Declarations

Competing Interests

Author Contributions

Data Availability

The authors declare there are no competing interests.

Caiyun Zhu conceived and designed the experiments, authored or reviewed drafts of the article, and approved the final draft.

Ran Xie performed the experiments, prepared figures and/or tables, and approved the final draft.

Fang Qin analyzed the data, prepared figures and/or tables, and approved the final draft.

Naiguo Wang performed the experiments, prepared figures and/or tables, and approved the final draft.

Hui Tang conceived and designed the experiments, authored or reviewed drafts of the article, and approved the final draft.

The following information was supplied regarding data availability:

This is a systematic review/meta-analysis and did not utilize original raw data.

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
