# Peer review of "Comparison of the recovery quality between remimazolam and propofol after general anesthesia: systematic review and a meta-analysis of randomized controlled trials"

_PeerJ, doi:10.7717/peerj.17930_

## Round 0.1 · original submission · Major Revisions

Thank you for submitting the manuscript. After evaluation, it needs adjustments, as specifically pointed out by reviewer #1. I would like clarification regarding the narrow time frame for the search of articles. Why not include a broader sample effort? The introduction is very general and does not present the state of the art in detail.

Reviewer 1 ·

Basic reporting

The authors do not clearly explain the rationale for comparing remimazolam and propofol. More context is needed on why this comparison is novel and addresses an important knowledge gap.
 The authors mention a previous meta-analysis that found no significant difference between the two drugs on the QoR-15 scale based on only two studies. It is questionable whether conducting another meta-analysis is necessary or meaningful given the continued lack of sufficient evidence.

Experimental design

Details are missing on the search strategy used for at least one database, making it difficult to assess the comprehensiveness of the literature search. The Embase database was not searched and should have been included.
 The inclusion criteria are poorly defined, e.g. it is unclear if cardiac surgery patients or those receiving nerve blocks were included. This lack of specificity undermines the robustness of the study design.
 The authors collected data on both the QoR-15 and QoR-40 scales but analyzed them separately without justification, rather than pooling the data.

Validity of the findings

Two of the included studies (refs 16 & 17) were published in Chinese and may not be identifiable in the searched databases, raising concerns about the validity of their inclusion.
 With only four studies available for the primary QoR-15 meta-analysis, the strength of evidence is low and may not be sufficiently reliable or interesting to readers to warrant publication.
 In summary, significant flaws are present in the rationale, methodology and interpretation that cast doubts on the validity and value of this meta-analysis in its current form.

·

Basic reporting

Minor revision requirieds

Experimental design

no comment

Validity of the findings

no comment

Additional comments

Minor revision requirieds

Reviewer 3 ·

Basic reporting

# Review of manuscripts :Comparison of the recovery quality between remimazolam and propofol after general anesthesia: systematic review and a meta-analysis of randomized controlled trials

The author described about the efficacy ofcomparison of the recovery quality between remimazolam and propofol after general anesthesia: systematic review and a meta-analysis of randomized controlled trials.

In this manuscripts,authers have shown that the recovery quality
of the remimazolam group after general anaesthesia was more superior. The incidence of
adverse events was low in remimazolam group.

While the results of the authors' study seem interesting, And the paper is well written, so I will only point out a few minor corrections.

Major comments

#As the authors point out, most of the papers collected in this meta-analysis were results of RCTs in Asian population. I think this is an important limitation.

Are there any data showing differences in intravenous anesthetics by race, not limited to this paper?

If so, please add them to the discussion section.

Experimental design

I think there is no concern about their experimental design.

Validity of the findings

This study has meaningful validity of the findings.

---

## Round 0.2 · accepted · Accept

After a careful review of the revised version of the manuscript, I confirm that the authors have addressed all of the reviewers' comments. The revision was thorough, and the points raised were appropriately addressed.

Therefore, I consider that this manuscript is ready for publication.

·

Basic reporting

Accept submission

Experimental design

Accept submission

Validity of the findings

Accept submission

Additional comments

Accept submission

Reviewer 3 ·

Basic reporting

No concern about this revised manuscripts.

Experimental design

No concern about this revised manuscripts..

Validity of the findings

No concern about this revised manuscripts..

Additional comments

No another comments about this manuscripts.